# Ozonated Sunflower Oil Embedded within Spray-Dried Chitosan Microspheres Cross-Linked with Azelaic Acid as a Multicomponent Solid Form for Broad-Spectrum and Long-Lasting Antimicrobial Activity

**DOI:** 10.3390/pharmaceutics16040502

**Published:** 2024-04-06

**Authors:** Roberto Spogli, Caterina Faffa, Valeria Ambrogi, Vincenzo D’Alessandro, Gabriele Pastori

**Affiliations:** 1Prolabin & Tefarm Srl, via dell’Acciaio N°9, 06136 Perugia, Italy; caterina.faffa@prolabintefarm.com (C.F.); gabriele.pastori@prolabintefarm.com (G.P.); 2Department of Pharmaceutical Sciences, University of Perugia, Via del Liceo 1, 06123 Perugia, Italy; valeria.ambrogi@unipg.it; 3Akeso Srl, Piazzale Italia 279, 55100 Lucca, Italy; vdalessandro2@gmail.com

**Keywords:** chitosan microspheres, azelaic acid, cross-linking, ozonated oils, mesoporous silica, emulsion, spray drying, multicomponent solid microspheres, peroxides release, synergic antimicrobial properties

## Abstract

Multicomponent solid forms for the combined delivery of antimicrobials can improve formulation performance, especially for poorly soluble drugs, by enabling the modified release of the active ingredients to better meet therapeutic needs. Chitosan microspheres incorporating ozonated sunflower oil were prepared by a spray-drying method and using azelaic acid as a biocompatible cross-linker to improve the long time frame. Two methods were used to incorporate ozonated oil into microspheres during the atomization process: one based on the use of a surfactant to emulsify the oil and another using mesoporous silica as an oil absorbent. The encapsulation efficiency of the ozonated oil was evaluated by measuring the peroxide value in the microspheres, which showed an efficiency of 75.5–82.1%. The morphological aspects; particle size distribution; zeta potential; swelling; degradation time; and thermal, crystallographic and spectroscopic properties of the microspheres were analyzed. Azelaic acid release and peroxide formation over time were followed in in vitro analyses, which showed that ozonated oil embedded within chitosan microspheres cross-linked with azelaic acid is a valid system to obtain a sustained release of antimicrobials. In vitro tests showed that the microspheres exhibit synergistic antimicrobial activity against *P. aeruginosa*, *E. coli*, *S. aureus*, *C. albicans* and *A. brasiliensis*. This makes them ideal for use in the development of biomedical devices that require broad-spectrum and prolonged antimicrobial activity.

## 1. Introduction

Multidrug therapy involves the simultaneous or sequential administration of two or more drugs to obtain a larger therapeutic benefit. Unfortunately, multidrug therapy has low patient compliance, resulting in poor adherence to therapy. Thus, interest in fixed-dose combinations is growing. Multicomponent solid forms seem to be more promising than fixed-dose combinations as they not only have the possibility of formulating drug–drug combinations but also can solve some problems connected to poor drug solubility or improve the formulation performance, allowing a modified release of active ingredients to better meet the needs of the therapy.

Chitosan (CH) is a natural, nontoxic, mucoadhesive, biocompatible and biodegradable substance extensively used in pharmaceutical and biomedical technology; among its numerous uses, it has been widely described also for its ability to increase drug permeability [1]. Chitosan microspheres (CH-MS) were found to exhibit great potential as a drug delivery system and have been proposed both for controlled drug release and for improving the bioavailability of poorly water-soluble drugs [2,3,4]. Moreover, chitosan has shown antimicrobial and antibiofilm activity [5] and for this reason its use has recently been proposed for making dressings for the prevention of microbial infections [6]. However, there are a few drawbacks associated with chitosan, such as lack of stability and poor mechanical properties [7].

Chitosan microspheres with improved mechanical strength and a long time frame can be obtained by the cross-linking of chitosan with a cross-linker agent. Regulation of the cross-linking density of the polymer matrix allows for the achievement of the desired degree of swelling and erosion of the microspheres and, hence, the controlled release of an entrapped active ingredient. To perform chitosan cross-linking, various chemical cross-linking agents, such as ammonium carbamate, vanillin, glutaric acid, genipin, glutaraldehyde, DL-glyceraldehyde, epichlorohydrin, sodium tripolyphosphate, glutaraldehyde, formaldehyde and glyoxal, have been used [8,9,10,11,12,13,14]. Ionic cross-linking agents are more biocompatible compared to the molecules listed above and can be used to obtain pH-sensitive microspheres [15,16]. The dicarboxylic acids, such as malonic, succinic, glutaric, adipic, pimelic, suberic and azelaic ones, have been used to cross-link chitosan by an emulsion cross-linking technique to obtain chitosan microparticles for bovine serum albumin delivery in tissue engineering [17,18,19]. Among them, azelaic acid not only is a cross-linking agent but also is endowed with antimicrobial activity and is reputed to be a promising agent for dermatological applications [20].

The search for new antimicrobial agents to be used as an alternative or a complementary therapy to classical antibiotics has been increasing. New antimicrobial agents should avoid resistance while maintaining reasonable side effects. Among new emerging agents, ozonated oils have been subject of increasing scientific interest and a greater number of clinical applications [21,22].

The main types of advanced formulations used to increase ozonated oil activity and reduce negative effects are emulsions (with different droplet sizes) and encapsulation techniques such as microencapsulation [23,24], liposomes [25,26] and cyclodestrins [27].

Recently, the use of chitosan in association with ozonated oil has attracted interest as a synergic antibacterial drug in medicaments for the treatment of persistent endodontic pathogens [28]; for the dermal treatment of Leishmania major promastigote/amastigote through the use of chitosan-coated nanoemulsions of ozonated olive oil [29]; and as bacteriostatic chitosan-based films containing encapsulated nanocapsules of ozonated olive oil [30]; however, to the best of our knowledge, no chitosan microspheres produced by spray drying have ever been proposed as a multicomponent solid form for modifying the release of ozonated oils.

This paper describes the preparation of multicomponent solid microspheres made of chitosan reticulated by azelaic acid containing ozonated sunflower oil. Microspheres were obtained by two different methods: one in the presence of mesoporous silica and the other in the presence of a surfactant. Azelaic acid was chosen as the dicarboxylic cross-linking agent because it is a widely used raw material in medical devices and has antibacterial properties that complement those of ozonated oil and chitosan. Moreover, by adjusting the degree of cross-linking, the modified release of the loaded drug can be obtained [31,32]. Mesoporous silica, whose peculiar characteristic is a high surface area, was used as an excipient able to act as a support for the ozonated oil, allowing it to be used in the spray-drying procedure without the presence of a surfactant.

After characterization, the prepared microspheres were evaluated in vitro for peroxides content and azelaic acid release profile, degradation rate and antimicrobial properties.

## 2. Materials and Methods

### 2.1. Materials

Chitosan (CH), MW = 200 kDa, deacetylation degree 75–85%; ozonated sunflower oil (OZ) were purchased by Alidans, Pisa (Italy); cosmetic-grade azelaic acid (AZA) with >90% purity was purchased by Gamma Chimica, Milan (Italy); glacial acetic acid >99% purity; sodium hydroxide pellets; ethanol 96%; dichloromethane (CH_2_Cl_2_); cetyl trimethyl ammonium bromide (CTAB); Tween 80 (TW80); sodium acetate anhydrous 99% purity; iron sulphate solution >90% purity; starch weld (soluble starch); sodium thiosulfate purity 99%; potassium iodide purity 99%; N,N-diethyl-p-phenylenediamine sulphate salt (DPD) 98% purity; KBr-FTIR grade were purchased by Merck Life Science S.r.l. Milano, Italy, hydrogen peroxide (35% w/w) was purchased by Alfa Aesar Thermo Fisher Gmbh, Kandel Germany, ultrapure water (H_2_O, conductivity < 1 µS) was obtained by the reverse osmosis plant of Prolabin & Tefarm, Perugia (Italy).

### 2.2. Equipment 

Chitosan microspheres were obtained using a MOD 1C/5UDF-T (I.C.F. & WELKO S.p.A., Modena, Italy) spray dryer equipped with double fluid and centrifugal disc nozzle; water evaporation capacity 3–6 L/h; inlet temperature up to 350 °C; outlet temperature 60–150 °C; maximum air consumption 400 Nl/min at 6 atm; maximum turbine rotation speed 30,000 rpm; drying chamber dimensions 1 m × 1.2 m × 2.7 m. Samples were processed and collected through a high-performance cyclone separator made of AISI 304 stainless steel.

Emulsions and dispersions were obtained using a jacketed borosilicate glass reactor of 5 L capacity and temperature control accuracy of ±0.1 °C equipped with a mechanical teflon blade. 

The synthesis of mesoporous silica was carried out using a 200 L borosilicate glass reactor by Steroglass srl, Perugia, Italy. The reactor was equipped with a thermal heating system, a condenser, a pH control system and an inlet valve for inert gases. An Ultra Turrax IKA T25 digital (Avantor delivered by VWR, Milano, Italy) was used for homogenizations.

### 2.3. Material Preparation Procedures

#### 2.3.1. Production of Chitosan Microspheres Cross-Linked with Azelaic Acid (CH-MS)

A 5 L aqueous dispersion containing 1% (*w*/*w*) chitosan and 0.18% (*w*/*w*) azelaic acid (CH/AZA ratio = 85/15) was stirred in a glass reactor for 30 min at 55 °C. Absolute AcOH was then added to achieve a concentration of 0.7% (*w*/*w*), and the dispersion was additionally stirred for 15 h at 55 °C. The dispersion was subsequently heated to 65 °C and then dried using a spray dryer to obtain the microspheres cross-linked with azelaic acid (CH-MS). The conditions for atomization involved an inlet temperature range of 180–220 °C and an outlet temperature range of 80–105 °C. The feeding speed was set between 1.5 and 2.5 L/h, and the turbine rotation speed was 15,000 rpm. The product underwent sieving using an ultrasonic vibrating sieve with a 75 µm cut-off.

#### 2.3.2. Synthesis of Mesoporous Silica (Si@)

The mesoporous silica was synthesized using a sol-gel process with CTAB as the templating agent and TEOS as the silicon source, following the established literature procedures [33,34]. The resulting silica was calcined. Specifically, 200 g of CTAB was dissolved in 95 L of a 1 M NaOH solution. The solution was heated to 80 °C, and then 1 L of TEOS was added. The reaction mixture was stirred for 2 h at this temperature (pH = 12.17). The obtained silica was separated from the solution by centrifugation at 10,000 rpm. It was then washed twice with a 1:1 mixture of water and ethanol, followed by a final wash with water. The resulting product was pre-dried at 80 °C for 12 h and calcined in a muffle at 600 °C for 5 h to obtain the mesoporous silica (Si@).

#### 2.3.3. Incorporation of OZ in Si@

An amount of 1.00 g of ozonated oil (OZ) was dissolved in 10 mL of CH_2_Cl_2_ in a 100 mL Erlenmeyer flask under magnetic stirring. Then, 2.33 g of mesoporous silica (Si@) were added to the mixture, in order to have an OZ/Si@ ratio of 30/70. The dispersion was stirred for 1 h at room temperature (r.t.), and the solvent was evaporated at reduced pressure (15 mTor, 30 °C) until a free-flowing dry powder (Si@OZ) was obtained.

#### 2.3.4. Synthesis of Chitosan Microspheres Cross-Linked with Azelaic Acid and Incorporating Ozonated Oil (CH-MS-OZ)

The procedure was the same as in Section 2.3.1, except that a 2/1 solution of TW80/OZ was added to the CH/AZA mixture while it was held at 65 °C in order to have the following weight ratios: CH/AZA/OZ/TW80 = 85/15/15/30. The resulting dispersion was stirred for 10 min at 65 °C followed by a 10 min homogenization to obtain a stable emulsion, which was then used for the atomization process to yield CH-MS-OZ microspheres.

#### 2.3.5. Synthesis of Chitosan Microspheres Cross-Linked with Azelaic Acid and Incorporating Ozonated Oil Loaded on Mesoporous Silica (CH-MS-Si@OZ)

The procedure was the same as in Section 2.3.1 except that Si@OZ powder was added to the CH/AZA mixture while it was held at 65 °C to give the following weight ratios: CH/AZA/Si@OZ = 85/15/60. The resulting mixture was homogenized for 15 min and then used for the atomization process to obtain CH-MS-Si@OZ microspheres.

Composition percentages of all samples (CH-MS, Si@OZ, CH-MS-OZ and CH-MS-Si@OZ) are reported in Table 1. 

### 2.4. Characterization

Morphological features were obtained by a LEO 1525 field emission scanning electron microscope with an EDX Bruker probe.

Particle-size distribution was determined by a laser diffraction particle-size analyzer Malvern Mastersizer 2000 (Alfatest, Milano, Italy). The samples were prepared by dispersing them in ultrapure water 1% TW80 by stirring for 10 min and sonication for an additional 10 min.

Surface area was set by a BET Micrometrics Gemini VII (Alfatest, Milano, Italy) surface analyzer for porous materials based on nitrogen adsorption–desorption isotherms.

Zeta potential was measured using an Anton Paar Litesizer 500 particle analyzer (Anton Paar Italia srl, Rivoli, Italy). The mesoporous silica and mesoporous silica loaded with ozonated oil samples were prepared by suspending them in ultrapure water 1% TW80 by stirring for 10 min and sonication for an additional 10 min. The surface charge of the microspheres was determined in phosphate buffers (0.02 M) at pH 7.4; the samples were prepared by suspending them at 1% (*w*/*v*) in buffer by ultrasonication for 30 min.

FTIR spectra were recorded by using a Jasco 4600 FTIR spectrometer in the range of 450–4000 cm^−1^. Each spectrum was a mean of 100 scans. The samples were analyzed in tablet form, obtained by mixing KBr and 1% sample powder. A FTIR spectrum of pure KBr was always recorded and subtracted as background.

Thermogravimetric analysis (TGA) and differential thermal analysis (DTA) were performed in air or nitrogen with a ramp up of 10 °C/min by a Perkin Elmer STA 8000 simultaneous thermal analyzer (Perkin Elmer, Milano, Italy).

X-ray XRPD was performed with a Bruker D2 advance diffractometer (Bruker D2 PHASER 2nd generation, (Bruker Italia srl, Milano, Italy) operating at 30 kV and 10 mA, a step size of 0.020 2 q degrees/step and time of 1.00 s per step of 1.00 s using copper K-α (Cu K-α) radiation and a multistrip LYNXEYE SSD160 detector. DIFFRACTPLUS EVA V4.2.2 software was used to perform the manipulations.

#### 2.4.1. Peroxide Titration and Encapsulation Efficiency (EE)

Peroxide titration was performed according to the iodometric titration method described by Cirlini et al. [35]. Briefly, OZ (ca. 0.1 g) or an amount of sample (Si@OZ, CH-MS-OZ and CH-MS-Si@OZ) containing OZ in the range of 0.3–0.6 g was added to 20 mL of CH_2_Cl_2_/AcOH 2/3 (*v*/*v*) solution with vigorous stirring for 30 min to obtain a complete OZ solubilization (test sample). The test sample containing Si@ continued to appear turbid due to the insolubility of Si@ in the medium. Successively, 1 mL of potassium iodate-saturated solution was added, and the mixture was stirred at room temperature for 16 h in the dark. Finally, the samples were mixed with 20 mL ultrapure water and 1 mL starch (1% solution) and titrated with sodium thiosulfate 0.05 M solution. OZ-free samples (Si@ and CH-MS) were also analyzed for background subtraction.

The peroxide value (*PV*) (meq × kg^−1^) of each sample was determined using Equation (1):(1)$$PV=\frac{V\times c\times 1000}{w}$$
where *c* is the concentration expressed as normality (eq) of the titrating solution, *w* is the weight (g) of the sample and *V* is the titrating solution volume (mL). *PV* is defined as the milliequivalents number (meq) of reactive oxygen present in 1000 g of *OZ*.

The encapsulation efficiency (*EE*) of the microspheres is expressed as the percentage of *OZ* actually loaded on the microspheres relative to the initial *OZ* used for microencapsulation. The amounts of *OZ* were quantified by the *PV*. The *EE* was determined using Equation (2):(2)$$EE\%=\frac{Actual\;OZ\;measured\;as\;PV}{Initial\;OZ\;used\;expressed\;as\;PV}\times 100$$

#### 2.4.2. Peroxides Release Test from OZ-Containing Samples

Samples (Si@OZ, CH-MS-OZ, CH-MS-Si@OZ) were suspended in 20 mL of 0.5 M acetate buffer pH 4 containing 1% *w*/*w* of TW80 and left under magnetic stirring at 25 °C for 7 days. Aliquots of 1 mL were taken at determined times (1 h, 4 h, 7 h, 24 h, 48 h, 96 h, 144 h, 168 h) and filtered (with 0.45 µm porosity nylon filter), and then peroxides release was determined by the DPD method; see Appendix A for a detailed description (Appendix A) [36].

#### 2.4.3. Percentage of Swelling Ratio

The swelling capacity of the microspheres was determined by keeping accurately weighed unloaded microspheres (500 mg) in 50 mL of phosphate buffered saline (PBS, pH 7.4) at 25 °C for 6 h. The particles were then separated by centrifugation at 6000 rpm for 10 min and reweighed after carefully blotting off the excess liquid with a tissue paper.

The swelling ratio was determined by Equation (3):(3)$$Swelling\;\left(\%\right)=\frac{\left(w_{s}-w_{0}\right)}{w_{0}}\times 100$$
where *w_s_* is the weight of swollen microspheres (mg) and *w_0_* is the weight of initial dry microspheres (mg). Data are expressed as mean ± standard deviation of five experiments.

#### 2.4.4. Percentage of Microspheres Degradation

Degradation tests: microsphere samples were incubated in phosphate buffer pH 7.4 (0.1 g/100 mL) for 35 days in a climatic chamber at 37 °C and 75% RH. After ageing, the suspension was centrifuged at 11,500 rpm for 10’, and the pellet obtained was dried at 70 °C for 5 h and then weighed to determine the residual weight. Percentage degradation is expressed as 100—residual weight (%) and was determined by Equation (4):(4)$$Percentage\;of\;degradation\;\left(\%\right)=100-\frac{w_{f}}{w_{0}}\times 100$$
where *w_0_* is the weight of initial dry microspheres (mg) and *w_f_* is the residual weight of dry microspheres (mg). Data are expressed as mean ± standard deviation of five experiments.

#### 2.4.5. Release of AZA

Release of AZA is expressed as % of release compared to the theoretical loading. An amount of 1.00 g of material was dispersed in 1 mL of acetate buffer in a screw-capped glass bottle and then transferred to a climatic chamber at 37 °C and 75% RH for ageing. At various interval times, 1 mL aliquots of the supernatant were taken, filtered through a 0.45 µm porosity nylon filter, concentrated under reduced pressure, then derivatized with diazomethane. The dimethyl ester of AZA was quantified by GC using a validated method (see Appendix A).

#### 2.4.6. Antimicrobial Activity

Antimicrobial activity was assessed by contaminating the samples with a defined inoculum of selected microbial strains and evaluating the reduction in contamination. The strains and inoculum concentrations used were as follows: *Pseudomonas aeruginosa* WDCM 00025 (2800–13000 CFU total), *Escherichia coli* WDCM 00012 (2300–11000 CFU total), *Staphylococcus aureus* WDCM 00032 (200–830 UFC), *Candida albicans* WDCM 00054 (2200–9100 CFU total), *Aspergillus brasiliensis* WDCM 00053 (48–1900 CFU total). The procedure involved dispersing 1 g of the tested samples in 10 mL of peptone solution, resulting in a suspension to which the pellet of the individual microbial strain was added. The resulting suspensions were stored at room temperature until inoculation, then 1 mL of each suspension was seeded into a Petri dish containing the selective agarose culture medium. All plates were incubated in thermostats for the necessary time to allow sufficient growth of the microorganisms for counting (*C. albicans* and *A. brasiliensis* 5 days; *P. aeruginosa* and *S. aureus* 48 h; *E. coli* 24 h). The number of colonies observed was corrected for the dilution factor to give CFU (colony forming units) per gram of product.

## 3. Results and Discussion

Two methods were used to load the ozonated sunflower oil (OZ) into the microspheres: in the first, the OZ was preloaded in a mesoporous silica (Si@) and then dispersed in the suspension with chitosan (CH) and azelaic acid (AZA) prior to spray drying; in the second, OZ embedding microspheres were obtained by spray-drying an O/W emulsion of OZ (O) in chitosan–azelaic acid dispersion (W) using TW80 as the emulsifier. Microspheres consisting of CH and AZA only were also prepared by spray drying. They were evaluated as a reference (Graphic Abstract).

The procedure used to cross-link chitosan and a dicarboxylic acid prior to spray drying (CH/AZA ratio = 85/15 and 15 h at 50 °C) was slightly modified from that used by Sedyakina with a view to promoting cross-linking [18].

The visual aspect and color of the obtained samples is shown in Figure 1. All samples containing chitosan are yellow and look like a dry powder except the material prepared in emulsion, which is oilier due to the TW effect. The Si@OZ is white and has good flowability. The materials do not have the typical rancid odor associated with OZ.

In Table 2, the main characteristics obtained from chemical–physical analyses are reported. As expected, Si@ had a high surface area, typical of mesoporous silica. The absorption of OZ into the mesopores was confirmed by the significant decrease in surface area from 773.3 m^2^/g to 182.4 m^2^/g and by the increase in particle size with a d_90_ varying from 3.61 µm of pristine material to 17.80 µm of Si@OZ The zeta potential changed slightly in the charged mesoporous silica from −20.0 mV for Si@ to −23.2 mV for Si@OZ, indicating that it is possible to disperse Si@OZ in aqueous solution in a sufficiently stable manner to allow spray drying.

Laser diffraction measurements indicate that the average particle-size distribution (PSD) of spray-dried microspheres was approximately between 10.61 and 38.87 μm. The use of AZA as a cross-linker produced particles with a narrow distribution between 4.40 and 20.76 μm. The inclusion of OZ, both in the case of Si@OZ and in the case of the O/W emulsion with TW80, resulted in larger particle sizes and a wider distribution of particles. The largest microspheres were obtained using the emulsion method.

The swelling values obtained partly reflect the CH% wt within the microspheres, which was highest for CH-MS (CH = 85% wt) and lowest for CH-MS-OZ (CH = 58.60% wt) and CH-MS-Si@OZ (CH = 53.12% wt). The CH-MS swelling of 379% is in good agreement with that found by Sedyakina for AZA cross-linked chitosan microparticles (about 450%); the lower degree of swelling observed can be attributed to the cross-link-promoting procedure employed in this study [18]. The swelling values decreased in both systems containing OZ, indicating that the destruction of hydrogen bonds within the chitosan matrix and the freedom to change the conformations of the macromolecules is limited, probably due to the more hydrophobic nature imparted by the presence of OZ.

The zeta potentials of the microspheres were measured in PBS at pH 7.4 and found to be positive, agreeing with the previous literature references. This indicates that only a part of the amino groups was neutralized during microsphere formation. The residual amino groups would be responsible for the positive zeta potential. The zeta potential was higher for CH-MS than for the other microspheres. This is probably due to the higher density of amine groups on the surface.

Microsphere degradation tests show that in all cases, a residue was present after 35 days of incubation. For CH-MS and CH-MS-OZ, the residue was 18.3% and 24.5%, respectively, probably due to the more cross-linked component. In the case of CH-MS-Si@OZ, a higher residue was observed due to the presence of insoluble Si@.

The expected sample composition was confirmed by FT-IR characterization. As represented in Figure 2a, the Si@OZ spectrum reports a combination of the main characteristic peaks of the species OZ and pure Si@, confirming the presence of both components in the sample. Peaks attributed to OZ are at 3010 cm^−1^ (cis –CH– stretching), 2926 and 2854 cm^−1^ (symmetrical and symmetrical aliphatic –CH_2_– stretching), 1748 cm^−1^ (carbonilic ester), 1460 cm^−1^ (aliphatic –CH_2_– and -CH_3_- bending), 1238 cm^−1^ and 1162 cm^−1^ (ester –CO– stretching) and 722 cm^−1^ (–CH_2_– rocking and out-of-plane vibrations of cis-replaced olephins overlapping). As concerns Si@, the main absorptions were at 1116 cm^−1^ (asymmetrical stretching of –Si–O–Si– bond) and 475 cm^−1^ (Si–O–Si bending in Si@). As depicted in Figure 2b, the AZA spectrum shows typical peaks of the aliphatic chain stretching vibrations at 2935 cm^−1^ (–CH–) and 2855 cm^−1^ (–CH_2_–), as well as a high-intensity peak at 1700 cm^−1^, due to the stretching vibrations of the two free carboxyl groups at the ends of the molecule.

The pure CH spectrum shows strong broad absorption at 3444 cm^−1^ attributed to OH and NH_2_ stretching vibrations, typical peaks of the aliphatic chain stretching vibrations at 2931 cm^−1^ and 2981 cm^−1^, of the amide bonds of the acylated monomers at 1657 cm^−1^ and at 1557 cm^−1^, of the free amine at 1155 cm^−1^ and at 1067 cm^−1^ corresponding to the glycosidic bond between the chitosan monomers and at 890 cm^−1^, typical of the glucopyranose ring.

The CH–MS spectrum shows a cross-linking between the amino groups of the glucopyranoside units and the carboxyl groups of azelaic acid with the increase in the intensity of the stretching vibrations of the amide bonds at 1557 cm^−1^. The incorporation of azelaic acid into the polymer matrix is also confirmed by the increased intensity of the bands of -CH– and –CH_2_– vibrations. Still, typical bands of amine and carboxylic groups, as well as those of glycosidic units and glucopyranoside rings, are still visible, suggesting that some azelaic acid molecules are not cross-linked to CH but probably are bound to the polymer skeleton by grafting [32].

Figure 2c shows that CH–MS–Si@OZ spectrum is totally superimposable tothe CH–MS one, except for three bands at 1748 cm^−1^ (OZ), 800 cm^−1^ (Si@) and 475 cm^−1^ (Si@), confirming the presence of Si@ and OZ in the sample. In the CH–MS–OZ spectrum, it is possible to observe an additional peak at 1748 cm^−1^ due to the OZ. New bands were observed in the 1717–1748 cm^−1^ range due to the C=O stretching vibrations of the carboxylic acid moiety for the cross-linked samples, in agreement with Sedyakina [18].

In order to evaluate the differences in shape and surface microstructures of the microspheres, SEM images were obtained, as shown in Figure 3. Images were captured with the scale bar equal to 20 µm and 2 µm. As shown in Figure 3a, CH–MS microspheres produced by CH/AZA = 85/15 exhibited a spherical structure with a smooth and nonporous surface. The perfect sphericity of the microspheres was probably favored by the chitosan–azelaic acid cross-linking. The microcapsules were distributed mostly in the size range of 2–20 μm in diameter, which is similar to that measured by laser diffraction particle-size analyzer. As shown in Figure 3b, CH–MS–OZ microspheres produced by CH/AZA/TW/OZ = 85/15/30/15 showed a less-than-perfect spherical structure with a smooth and nonporous surface. Their macroscopic morphology indicated that the presence of TW/OZ influenced the spherical structure and the size, which was larger than the CH–MS microspheres. Within the morphological distribution, smaller microspheres merged with larger ones to form clusters. As shown in Figure 3c, the CH–MS-Si@OZ microspheres prepared from CH/AZA/Si@OZ = 85/15/60 showed a predominant population consisting of particles with an irregular, wrinkled, pseudospheric morphology composed of a chitosan matrix containing aggregated silica particles and a minority population of smooth spherical microspheres presumably containing silica. The size of these particles ranged from 2 to 50 μm.

Thermogravimetric profiles in air are reported in Figure 4. The theoretical loading of 30% *w*/*w* oil in mesoporous silica was confirmed by the 28.9% weight loss of the Si@OZ sample, since the mesoporous silica Si@ had a neglectable weight loss up to 800 °C. For CH–MS and CH–MS–OZ, the total mass lost at 600 °C was almost total, as expected for an organic compound with a residue, respectively, of 5.6% and 9.7%.

The thermogram of CH–MS–Si@OZ, with a residue at 600 °C of 29.9% wt, confirmed the incorporation of Si@, which theorical content in this sample is 26.5% (the data is reported in Table 1).

The PV indicates the amount of peroxide in a sample expressed as the active oxygen amount per kilogram of sample (meq/kg) [37]. This method is used to assess the stability of ozonated vegetable oil and to check its storage conditions but also to correlate its potential antimicrobial activity [38].

Table 3 shows the measured PV of the microspheres and Si@OZ according to the method of Cirlini [35]. The percentage composition value of OZ in the samples was calculated proportionally comparing the measured value of PV of the samples and the PV of pure OZ (set as 100%). The percentage ratio between the measured and theoretical % OZ gives the encapsulation efficiency (EE), which was between 75.7% and 82.1% for all samples. In the case of the microspheres, these data indicate that although OZ is a material that can undergo thermal degradation [39], the microencapsulation process used by spray drying does not significantly alter its stability. The higher EE value for the Si@ embedding system is probably due to the protection provided by the inorganic solid against thermo-oxidative processes.

The DTA curve of Si@OZ in Figure 5 shows the typical endothermic peaks associated with the thermo-oxidative degradation of OZ [40], peaks that are also found in OZ-containing microspheres. In the CH curve, the typical endothermic peak appears at around 620 °C. This peak is clearly detectable in all microspheres, but it is shifted to lower temperatures, indicating that the microspheres had a lower thermal stability than pure chitosan. In the DTAs, the AZA melting peak at 108.57 °C is not visible, indicating that AZA was not present in crystalline form. This finding was also confirmed by XRPD analysis in Figure 6, where AZA was not detectable in crystalline form, confirming that it was molecularly dispersed or bound to the chitosan in the composite. XRPD diffractograms showed that the reference chitosan was more crystalline than the spray-dried microspheres and confirmed that the mesoporous silica was amorphous.

The amount of ozonide incorporated into OZ is an important measure of the oil’s ability to provide active O_2_ and other active species that can be exploited in the treatment of skin diseases [37]. It follows that knowledge of the rate of formation of peroxide species is useful in achieving the desired therapeutic effects. However, research into the antimicrobial activity of these products has been hampered by the lack of standardized and reliable in vitro screening methods [21].

The quality of the ozonated oils obtained can be assessed by measuring a number of physicochemical parameters, including PV. However, the amount of peroxide products formed in in vitro tests is very small, and the currently available methods, including PV, are not sensitive or measurable enough to monitor them over time.

OZ is a species which is not directly detectable through a spectrophotometric method; thus, its concentration was determined indirectly, on account of the oxidative properties of the ozonated compound present in OZ. As a model system to test the DPD method, hydrogen peroxide (H_2_O_2_) was selected. Whichever oxidative species are present in OZ, it is possible to quantify their oxidative properties by the DPD method and express the results as millimoles of peroxides.

DPD is oxidized by H_2_O_2_ with the generation of DPD^•+^ radical, which shows a maximum absorbance of light at 553 nm (A553). Under optimal conditions (pH 4; 20 mM DPD; 1. mM FeSO_4_ and reaction time of 45 s), the increase in A553 is linear with the [H_2_O_2_]; see Appendix A for more information (Appendix A).

The release study was carried out using 0.5 M acetate buffer pH 4 with 1% TW80 as the release medium. The choice to perform the releases at pH 4.0 instead of pH 7.4 was made to promote the stability of the peroxides in view of the prolonged measurement time [41]. The concentration of the samples in the release medium was adjusted so that each sample had the same PV of 300 millimoles of peroxide in 20 mL of medium at the start of this study, which corresponds to a concentration of approximately 0.5 g of sample in 20 mL of medium.

The OZ curve shows an increasing release of peroxide species over time, and in absolute value, this is greater than for the microencapsulated systems (Figure 7). The rate of peroxide formation by the CH–MS–OZ and Si@OZ systems is very low compared to OZ, but an increasing trend over time can also be seen. For the CH–MS–OZ microspheres, a rate of peroxide formation superimposed on that of OZ was observed up to 48 h, then slowing down and remaining constant up to seven days; this behavior can be explained by the degradation data of the microspheres, considering that they disintegrated rapidly in the first 48 h, while the remaining part, probably the cross-linked part, generated the prolonged release of oxidizing species.

As a general observation, it can be said that microencapsulation slows down the formation of peroxides compared to OZ and that mesoporous silica is the one that achieves a greater slowing effect among the systems studied.

The study of AZA release from microspheres was carried out in phosphate buffer at 37 °C and 75% RH, followed by determination of AZA by GC after derivatization with diazomethane to form the AZA dimethyl ester (see Appendix A for more information). A rapid initial AZA release was observed (Figure 8), reaching 33.5% after 24 h, followed by a slower subsequent progression to 80% after 6 days. Analysis after 35 days showed that the percentage of released AZA remained stable at 80%, indicating, in good agreement with the degradation data (Table 2), that a part of the AZA remained insoluble, probably as a consequence of high cross-linking with CH.

The antimicrobial tests showed that all samples had an antimicrobial activity, albeit with different performances both in terms of specificity towards the different microbial strains tested and in terms of intensity and duration of action (Table 4 and Figure 9a–c). CH-MS totally inhibited the growth of the bacteria *E. coli* and *S. aureus*, while it inhibited the growth of *P. aeruginosa* by 60%. Si@OZ totally inhibited the growth of *P. aeruginosa* bacteria, with increasing inhibition over time towards *E. Coli* with 12% knockdown after 10 min; 37% after 22; and 44% after 4 h and 67% knockdown for *S. Aureus* after 2 and 4 h. C-MS-OZ totally inhibited the bacterial growth against all bacterial strains tested; C-MS-Si@OZ completely inhibited the growth of the bacteria *E. coli*, with an increasing reduction over time of up to 92% at 4 h for *P. aeruginosa* and about 80% for S. aureus.

Taken together, these data indicate that all microcapsules containing CH and AZA were very effective in completely inhibiting the growth of *E. coli* after 10 min. The OZ-containing materials showed the same efficacy against *P. aeruginosa*. The CH-MS microspheres showed equally effective inhibition against *S. Aureus*.

Interestingly, the CH-MS-OZ and CH-MS-Si@OZ hybrid systems combined the best antimicrobial properties of the individual components. In particular, the CH-MS-OZ system was effective against all bacterial strains after 10 min, whereas the CH-MS-Si@OZ microspheres were slightly less effective because the release of OZ and the hydrolysis of the microsphere were slowed down in this composite.

As reported in Table 5 and shown in Figure 10, CH-MS showed total growth inhibition against *C. albicans* after 30 min and up to 48 h and partial inhibition between 30 and 69% in the case of *A. brasiliensis*. Si@OZ showed a total inhibition of *C. albicans* growth after 15 h, while for *A. brasiliensis*, an inhibition of about 60% was observed between 4 and 24 h, reaching 96% after 48 h.

CH-MS-Si@OZ showed an inhibition against *C. albicans* of 88–72% in the first 24 h, decreasing to 25% at 48 h, and an increasing knockdown over time against *A. brasiliensis*, reaching 83% at 48 h. CH-MS-OZ showed a total inhibition against *C. albicans* after 15 h and an inhibition of 88–70% against *A. brasiliensis* between 15 and 48 h.

In general, therefore, it can be observed that chitosan microspheres are most effective in inhibiting the growth of *C. albicans*, whereas OZ- encapsulating materials are most effective against *A. brasiliensis*. Microspheres combining chitosan, azelaic acid and OZ showed inhibitory activity against the two fungal strains, with CH-MS-OZ in particular optimizing and balancing the best antimicrobial performance of the individual actives.

## 4. Conclusions

In the present study, OZ embedding chitosan microspheres cross-linked with azelaic acid were prepared for the first time, and it was shown that the spray-drying production method can be used without damaging the peroxide content of the ozonated oil.

An analytical method with micromolar sensitivity was developed to indirectly monitor the release of ozonated oil by detecting peroxides produced by the reaction between ozonides and water.

The multiphase solid formulations obtained were shown to be extremely effective in inhibiting the growth of multiple strains of bacteria and fungi; indeed, the combined and sustained release of antimicrobial actives showed a broader and more prolonged inhibitory effect compared to the action of any individual actives. Moreover, they address the technical issues currently faced in the therapeutic use of ozonated oils, such as unpleasant odor, high viscosity and adverse effects on the skin caused by overly aggressive action.

Based on the results obtained, the developed materials could be used in the manufacture of biomedical devices to achieve a broad-spectrum and prolonged antimicrobial effect. In particular, the microspheres could be incorporated in advanced dressings based on resorbable biopolymers or cellulose fibers, in bioscaffolds for dermal or bone regeneration and in formulations for the treatment of mucous membranes such as nasal, gingival or vaginal.

## 5. Patents

This work has led to the submission of the following Italian patent applications:(1)N°102023000013839—priority 04/07/23, Title: Cosmetic composition containing microspheres of chitosan and azelaic acid functionalized with ozonated oil.(2)N°102023000013845—priority 04/07/23, Title: Cosmetic composition containing chitosan microspheres and azelaic acid.(3)N°102023000013838—priority 04/07/23, Title: Cosmetic display containing chitosan microspheres and azelaic acid.(4)N°102023000013872—priority 04/07/23, Title: Advanced medication in the form of a collagen sheet containing chitosan microspheres and azelaic acid functionalized with ozonated oil loaded on mesoporous silica.(5)N°102023000013839—priority 04/07/23, Title: Chitosan and azelaic acid microspheres functionalized with ozonated oil loaded on mesoporous silica.(6)N°102023000013890—priority 04/07/23, Title: Topical composition containing silica functionalized with oil for wound debridement.(7)N°102023000013899—priority 04/07/23, Title: Advanced medication containing chitosan microspheres and azelaic acid functionalized with ozonated oil.

## Figures and Tables

**Figure 1 pharmaceutics-16-00502-f001:**
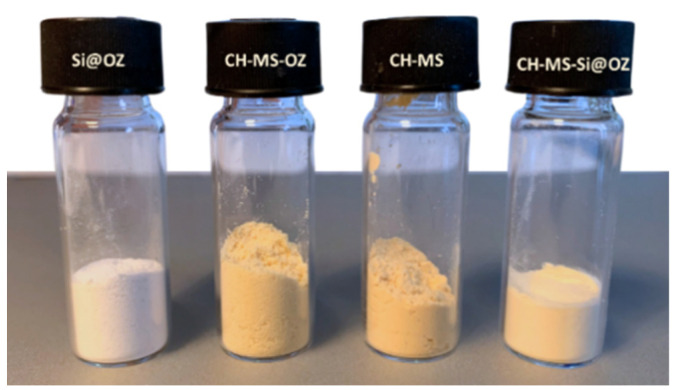
Visual aspect, from the left to the right, of Si@OZ, CH-MS-OZ, CH-MS, CH-MS-Si@OZ.

**Figure 2 pharmaceutics-16-00502-f002:**
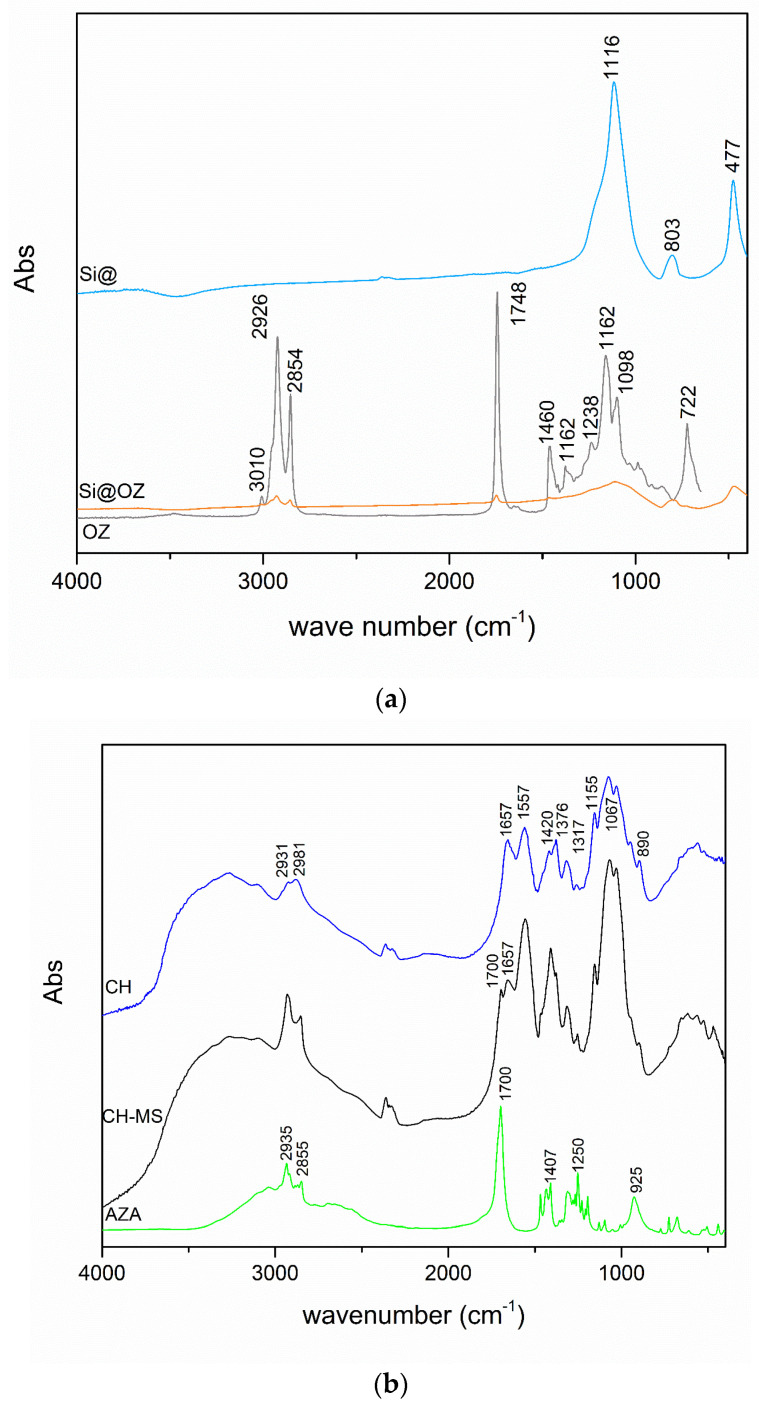
(**a**) FT-IR spectra of Si@, OZ, Si@OZ; (**b**) FT-IR spectra of CH, CH-MS, AZA; (**c**) FT-IR spectra of CH-MS-OZ, CH-MS, CH-MS-Si@OZ.

**Figure 3 pharmaceutics-16-00502-f003:**
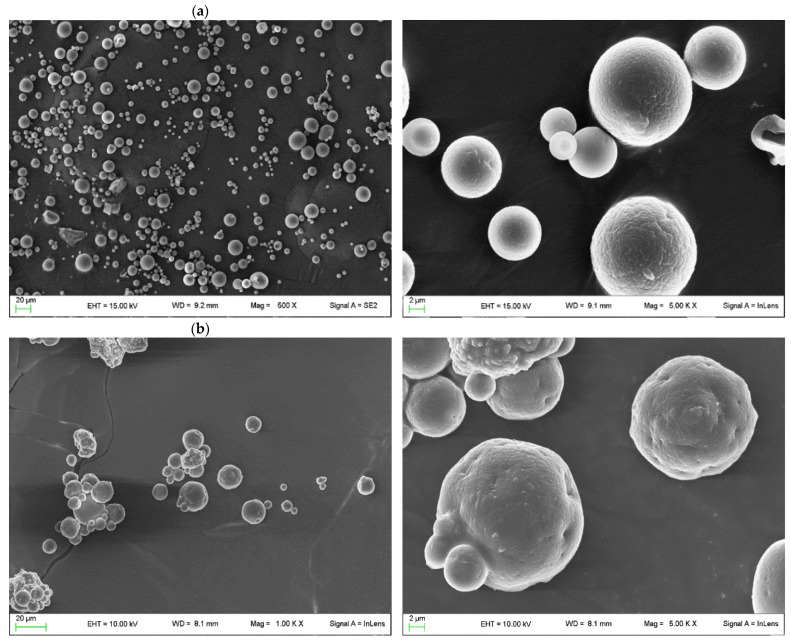
SEM pictures of samples (**a**) CH–MS, (**b**) CH–MS–OZ and (**c**) CH–MS–Si@OZ.

**Figure 4 pharmaceutics-16-00502-f004:**
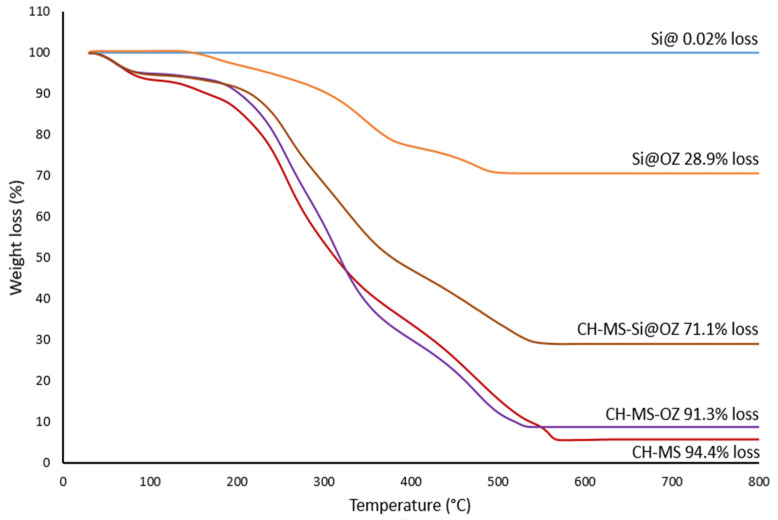
TGA profiles and residues.

**Figure 5 pharmaceutics-16-00502-f005:**
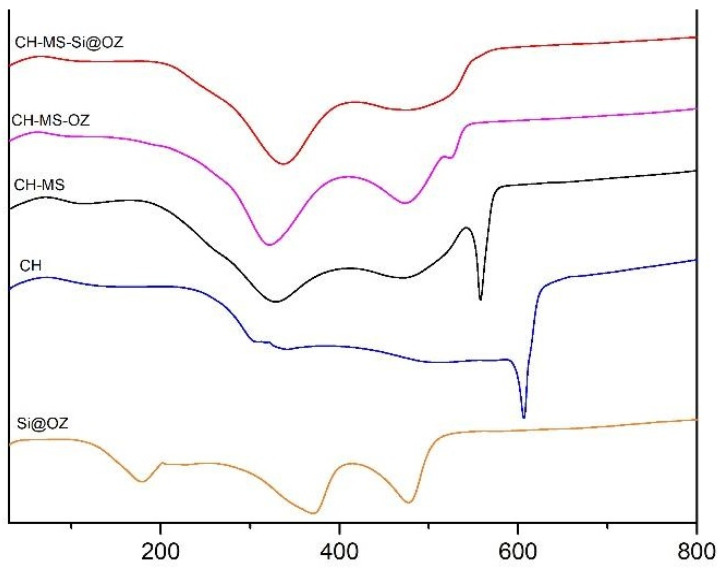
DTA curves in air atmosphere for Si@OZ, chitosan, CH-MS, CH-MS-OZ, CH-MS-Si@OZ.

**Figure 6 pharmaceutics-16-00502-f006:**
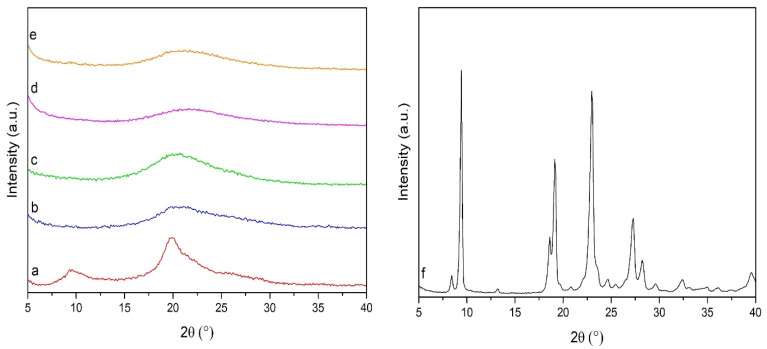
XRPD patterns of (**a**) chitosan, (**b**) CH-MS, (**c**) CH–MS–OZ, (**d**) Si@OZ, (**e**) CH-MS-Si@OZ, (**f**) AZA.

**Figure 7 pharmaceutics-16-00502-f007:**
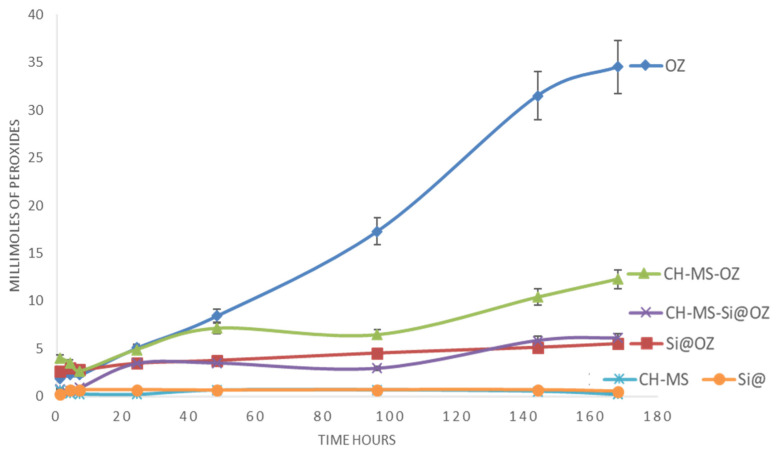
OZ release in acetate buffer pH 4 measured as millimoles of peroxides measured at various times.

**Figure 8 pharmaceutics-16-00502-f008:**
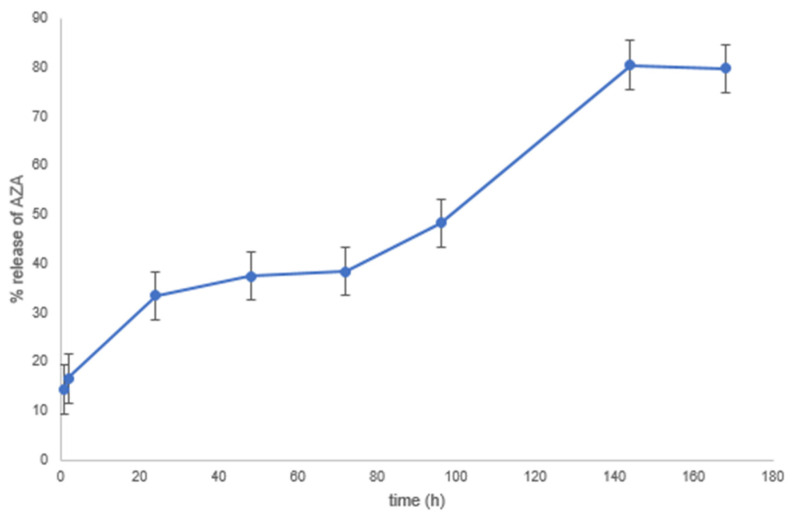
In vitro drug release kinetics of AZA from CH-MS in phosphate buffer at 37 °C.

**Figure 9 pharmaceutics-16-00502-f009:**
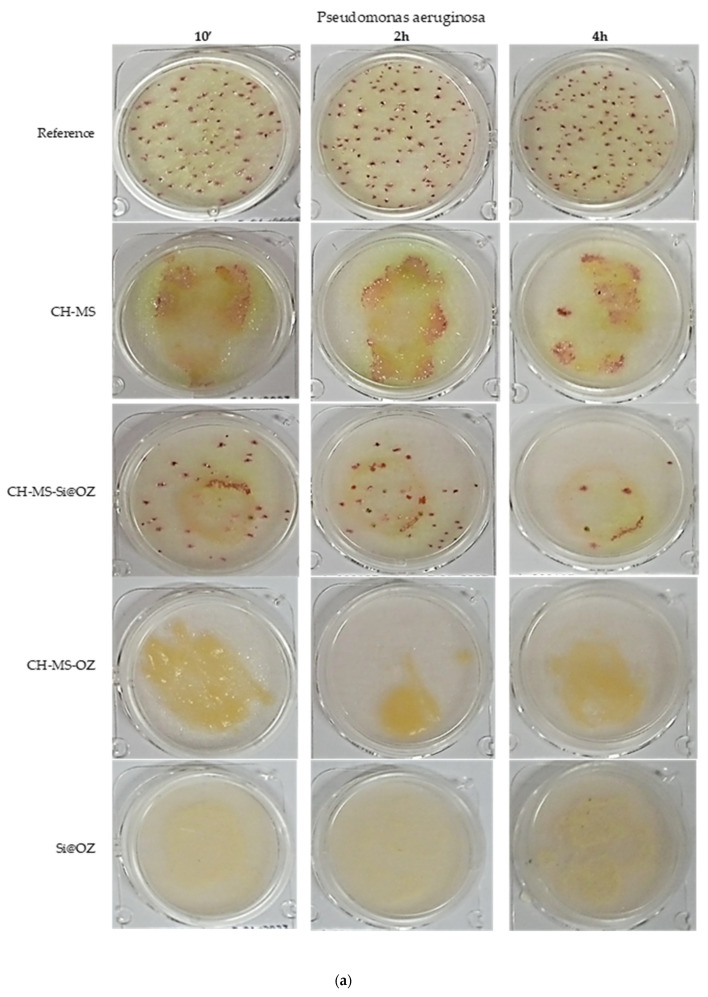
(**a**) Culture plates showing antimicrobial activity over time for *P. aeruginosa* strains. (**b**) Culture plates showing antimicrobial activity over time for *E. coli* strains. (**c**) Culture plates showing antimicrobial activity over time for *S. aureus* strains.

**Figure 10 pharmaceutics-16-00502-f010:**
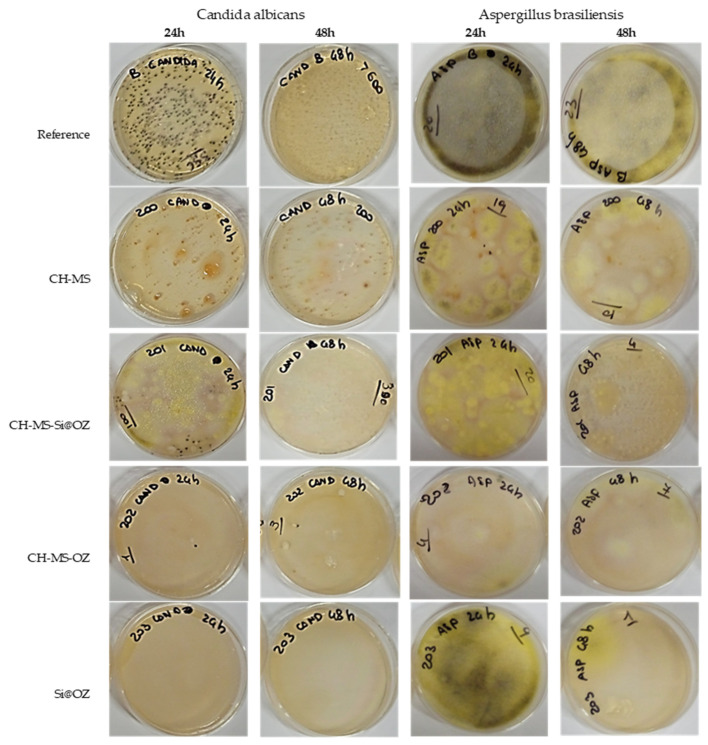
Culture plates showing antimicrobial activity over time for fungal strains.

**Table 1 pharmaceutics-16-00502-t001:** Composition of the mixture for microsphere preparation.

	CH (%)	AZA (%)	Si@ (%)	OZ (%)	TW80 (%)
CH-MS	85	15	-	-	-
Si@OZ	-	-	70	30	-
CH-MS-OZ	58.60	10.35	-	10.35	20.70
CH-MS-Si@OZ	53.12	9.37	26.25	11.25	-

**Table 2 pharmaceutics-16-00502-t002:** PSD, surface area, zeta potential, swelling and degradation (35 days, PBS, 37 °C) of prepared samples.

	D_10_	d_50_	d_90_	Surface Area m^2^/g	Zeta Potential mV	Swelling %	Degradation %
	µm						
Si@	0.54	0.12	3.61	773.2	−20.0 ± 0.5	-	-
Si@OZ	1.25	3.91	17.80	182.4	−23.2 ± 0.3	-	-
CH-MS	4.40	10.61	20.76	-	+26.4 ± 0.7	379 ± 12	81.7 ± 2.5
CH-MS-OZ	16.88	38.37	75.05	-	+18.0 ± 0.8	197 ± 25	75.5 ± 3.5
CH-MS-Si@OZ	4.84	26.92	52.10	-	+19.3 ± 0.5	150 ± 10	56.0 ± 1.9

**Table 3 pharmaceutics-16-00502-t003:** Peroxide value (PV), percentage of ozonated oil loaded in the analyzed samples calculated by PV (% OZ Calculated), theoretical PV for total load (% OZ Theoretical) and encapsulation efficiency (% EE).

	PV * meq × kg^−1^	% OZ Calculated	% OZ Theoretical	% EE
OZ	2156 ± 7.2%	100.0	-	-
Si@OZ	524 ± 14.3%	24.3	30.0	81.0
CH-MS-OZ	169 ± 7.8%	7.8	10.3	75.7
CH-MS-Si@OZ	198 ± 10.5%	9.2	11.2	82.1

* Average of 5 measurements.

**Table 4 pharmaceutics-16-00502-t004:** Antimicrobial activity and % inhibition over time for bacterial strains.

Sample	UFC 10′	% Inhibition	UFC 2 h	% Inhibition	UFC 4 h	% Inhibition
*Pseudomonas aeruginosa*						
Reference	91	0%	107	0%	101	0%
CH-MS	30	67%	40	63%	40	60%
CH-MS-Si@OZ	40	56%	33	69%	8	92%
C-MS-OZ	0	100%	0	100%	0	100%
Si@OZ	0	100%	0	100%	0	100%
*Escherichia coli*						
Reference	182	0%	150	0%	233	0%
CH-MS	0	100%	0	100%	0	100%
CH-MS-Si@OZ	1	99%	2	99%	0	100%
CH-MS-OZ	0	100%	0	100%	0	100%
Si@OOZ	161	12%	94	37%	131	44%
*Staphylococcus aureus*						
Reference	31	0%	41	0%	45	0%
CH-MS	0	100%	0	100%	0	100%
CH-MS-Si@OZ	4	87%	12	71%	6	87%
CH-MS-OOZ	0	100%	0	100%	0	100%
Si@OZ	20	35%	14	66%	15	67%

**Table 5 pharmaceutics-16-00502-t005:** Antimicrobial activity and % inhibition over time for fungal strains.

	UFC 30′	% Inh *	UFC 4 h	% Inh *	UFC 15 h	% Inh *	UFC 24 h	% Inh *	UFC 48 h	% Inh *
*Candida albicans*										
Reference	242	0%	200	0%	200	0%	356	0%	>400	0%
C-MS	1	100%	8	96%	1	100%	0	100%	0	100%
C-MS-Si@OOZ	30	88%	28	86%	50	75%	100	72%	300	25%
C-MS-OOZ	31	87%	8	96%	0	100%	1	100%	3	99%
Si@OOZ	133	45%	21	90%	0	100%	0	100%	0	100%
*Aspergillus brasiliensis*										
Reference	32	0%	22	0%	24	0%	22	0%	23	0%
C-MS	10	69%	13	41%	15	38%	15	32%	10	57%
C-MS-Si@OOZ	11	66%	19	14%	16	33%	10	55%	4	83%
C-MS-OOZ	26	19%	17	23%	3	88%	4	82%	7	70%
Si@OOZ	28	13%	7	68%	8	67%	9	59%	1	96%

* Inhibition %.

## Data Availability

Data are contained within the article and Appendix A.

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
