# Peer review of "Ozonated Sunflower Oil Embedded within Spray-Dried Chitosan Microspheres Cross-Linked with Azelaic Acid as a Multicomponent Solid Form for Broad-Spectrum and Long-Lasting Antimicrobial Activity"

_pharmaceutics, 2024, doi:10.3390/pharmaceutics16040502_

Round 1
Reviewer 1 Report
Comments and Suggestions for Authors
This is well written paper which demonstrates formulation of stable drug delivery systems for ozonated sunflower oil with possible use in antimicrobial therapy. Experiments are well designed with adequate discussion of the obtained results. I have only some minor suggestions to the authors:
1. Span value is commonly stated in particle size analysis.
2. Zeta P. is very uncommon abbreviation for zeta potential.
3. Please discuss if there is potential shifting in peak positions in the FTIR spectra, not only intensity changes.
4. Due to huge differences in the peak intensities, PXRD patterns of pure API and formulations should be presented in separate plots.
5. Please check typing errors (Line 280, 352, table 1...)
Comments on the Quality of English LanguagePlease check some minor typing errors.
Author Response
Dear reviewer, thank you for your positive feedback on our work. We appreciate your suggestions and here are the answers to the highlighted considerations:
- Span value is commonly stated in particle size analysis.
Answer: European Pharmacopoeia suggests that particle size may be charactesized in the following manner:
x90 = particle size corresponding to 90 per cent of the cumulative undersize distribution;
x50 = median particle size (i.e. 50 per cent of the particles are smaller and 50 per cent of the particles are
larger);
x10 = particle size corresponding to 10 per cent of the cumulative undersize distribution.
Thus, we chose to use this way for describing the microparticles size.
- Zeta P. is very uncommon abbreviation for zeta potential.
Answer: we agree that this abbreviation is rarely used and could lead to misunderstandings. As the term only appears eight times in the document, we have decided to leave the diction in full.
- Please discuss if there is potential shifting in peak positions in the FTIR spectra, not only intensity
Answer: Closer analysis of the FT-IR spectra showed that new bands were present in the spectrum of the microspheres compared to those of the reagents. The following sentence has been added to the text (see line 357-359): “New bands were observed in the 1717-1748 cm-1 regions due to the C=O stretching vibrations of the carboxylic acid moiety for the crosslinked samples in agreement with Sedyakina [18].”
4 Due to huge differences in the peak intensities, PXRD patterns of pure API and formulations should be presented in separate plots.
Answer: The XRPD spectrum was spitted in two separate plot as suggested.
- Please check typing errors (Line 280, 352, table 1...)
Answer: The entire document has been carefully corrected for typographical and other errors.
Reviewer 2 Report
Comments and Suggestions for Authors
Comments to the Author:
In this work spray-dried chitosan microspheres cross-linked with azelaic acid and embedding ozonated sunflower oil are described. It was shown that the spray drying production method can be used without damaging the PV title of the ozonated oil. It was found that the systems obtained are multiphase solid formulations for the combined and sustained release of antimicrobial actives such as chitosan, azelaic acid and ozonated oil. The work is of interest because the materials produced could be used in the manufacture of biomedical devices to obtain a broad-spectrum and prolonged antimicrobial effect. The article looks like a Full Article and may be published after minor revision.
Notes:
1. The meaning of “PSD” abbreviation (Table 2) should be noted in the text.
2. The label axis in the Figure 4 should be increased for clarity.
3. Why OZ release was studied in acetate buffer at pH 4 (Figure 7)? Short comment should be added in the text.
4. Conclusions should be shortened and focused on the main findings of this work.
Author Response
Dear reviewer, thank you for your positive feedback on our work. We appreciate your suggestions for improvement.
Here are the answers to the highlighted considerations:
- The meaning of “PSD” abbreviation (Table 2) should be noted in the text.
Answer: the PDS meaning was inserted at line 294
- The label axis in the Figure 4 should be increased for clarity.
Answer: all images have been enlarged
- Why OZ release was studied in acetate buffer at pH 4 (Figure 7)? Short comment should be added in the text.
Answer: see the following sentence at line 451: The choice to perform the releases at pH 4.0 instead of pH 7.4 was made to promote the stability of the peroxides in view of the prolonged measurement time [41].
- Conclusions should be shortened and focused on the main findings of this work.
Answer: Conclusions have been shortened and focused on the main findings of the work.
Reviewer 3 Report
Comments and Suggestions for Authors
This article described a novel multicomponent delivery system combining chitosan microspheres, ozonated oil, and azelaic acid designed to achieve sustained release of antimicrobial drugs. The article details two methods of incorporating ozonized oil into microspheres and evaluates the encapsulation efficiency of ozonized oil. This is an interesting work.
However, there are several concerns:
1. Please check and correct the format of each reference carefully to ensure that it conforms to the journal's specifications.
2. For the details of the article, there are a few points that need attention.
1) The writing format of the title should be consistent, such as "2.3.1." should be changed to "2.3.1".
2) In 2.3.1, the second line of the “30 min. at 55 ℃.” There is a error, it should be “30 min at 55℃”. In addition, the font of “for” in the fourth line of the section is inconsistent, please make a uniform adjustment.
3) In Table 2, “-22..9” should be changed to “-22.9”.
4) In page 8, “MS-CH” should be changed to “CH-MS”.
5) In Table 3, the data 82.1 in column EE is missing the unit %.
3. In Table 2, why the loaded zeta potential becomes more negative indicates that the obtained system is sufficiently stable, it is recommended that an explanation be given in the article. If there are relevant references to support this, please cite them as well to enhance the persuasiveness of the article.
4. The scale bar mentioned in the article are “20 mm and 2 mm”, but shown as 20 μm and 2 μm in Figure 3a/b/c. Please verify and harmonize the scale bar to ensure that the descriptions in the article correspond to the actual scale bar in the figure.
5. As for the antimicrobial experiments, if possible, it would be more convincing to provide relevant pictures as illustrations. Pictures can visualize the results of the experiments.
Comments on the Quality of English LanguagePlease proof-read the manuscript again, as there are some typo errors.
Author Response
Dear reviewer, thank you for your positive feedback on our work. We appreciate your suggestions for improvement.
Here are the answers to the highlighted considerations:
- Please check and correct the format of each reference carefully to ensure that it conforms to the journal's specifications.
Answer: references were carefully checked and adjusted to the to the journal's specifications.
- For the details of the article, there are a few points that need attention.
1) The writing format of the title should be consistent, such as "2.3.1." should be changed to "2.3.1".
Answer: title writing format was adjusted and standardized throughout the document.
2) In 2.3.1, the second line of the “30 min. at 55 ℃.” There is a error, it should be “30 min at 55℃”. In addition, the font of “for” in the fourth line of the section is inconsistent, please make a uniform adjustment.
Answer: notations have been corrected and standardized throughout the document.
3) In Table 2, “-22..9” should be changed to “-22.9”.
Answer: The value of the zeta potential has been changed due to a review and corrected notation.
4) In page 8, “MS-CH” should be changed to “CH-MS”.
Answer: correction done
5) In Table 3, the data 82.1 in column EE is missing the unit %.
Answer: the unit of percentage has been reported in the title
- In Table 2, why the loaded zeta potential becomes more negative indicates that the obtained system is sufficiently stable, it is recommended that an explanation be given in the article. If there are relevant references to support this, please cite them as well to enhance the persuasiveness of the article.
Answer: Thank you for this note, in fact a re-examination of the data showed this to be unusual. Why should the mesoporous solid have a more negative potential as a result of the oil adsorption process? New zeta potential measurements were carried out to verify the experimental data. The new measurements confirmed the value of the zeta potential for the oil-loaded mesoporous solid and reduced the difference with the unloaded mesoporous silica to 3 mV. The solid loaded with ozonated oil still has a more negative zeta potential, but only slightly; we searched the literature for a precedent, but found none. The control of the measurements reassures us of the scientific accuracy of the data obtained.
- The scale bar mentioned in the article are “20 mm and 2 mm”, but shown as 20 μm and 2 μm in Figure 3a/b/c. Please verify and harmonize the scale bar to ensure that the descriptions in the article correspond to the actual scale bar in the figure.
Answer: notations have been corrected and standardized throughout the document.
- As for the antimicrobial experiments, if possible, it would be more convincing to provide relevant pictures as illustrations. Pictures can visualize the results of the experiments.
Answer: relevant pictures of the antimicrobial test were inserted visualize the results of the experiments.
Round 2
Reviewer 3 Report
Comments and Suggestions for Authors
This revised version can be acceptable.